# Verifying Meta-Awareness via Predictive Rewards in Reasoning Models

**Yoonjeon Kim** [* 1]   **Doohyuk Jang** [* 1]   **Eunho Yang** [1 2]

## Abstract

Recent research on reasoning models explores the meta-awareness of language models, including their ability to determine optimal thinking duration, recognize knowledge boundaries, and structure concept-level thinking. While current large reasoning models depend solely on answer-based verification, we show that adding meta-awareness objectives leads to significant performance gains over models without such meta-knowledge. **MAPR** (**M**eta-**A**wareness via **P**redictive **R**eward) utilizes a self-generated task of predicting rollout statistics - specifically length, pass-rate, and concepts used - allowing for verification against the actual statistics. Furthermore, by leveraging this self-predictive capability, the model can regulate its reasoning behavior by i) filtering out trivial or unsolvable prompts, ii) reducing lengthy generations that tend to be incorrect, and iii) generating hints relevant to the problem. The results are promising: **MAPR** yields significant improvements in both accuracy and training efficiency on various reasoning benchmarks. More specifically, our method can speed up GRPO training by over $1.28\times$ to reach the same performance, and achieve an 83.18% gain in accuracy on AIME25, and a 16.45% average gain over six mathematical benchmarks. The code is publicly available at https://github.com/akatigre/MAPR-RL.

## 1. Introduction

Recent studies have confirmed that applying RL-based post-training to large language models (LLMs) (Brown et al., 2020; Yang et al., 2025a; Touvron et al., 2023) can significantly enhance their reasoning ability. In particular, methods such as GRPO (Shao et al., 2024), which efficiently train

---
*Equal contribution [1]KAIST, Daejeon, South Korea [2]AITRICS, Seoul, South Korea. Correspondence to: Eunho Yang <eunhoy@kaist.ac.kr>.

*Proceedings of the $43^{rd}$ International Conference on Machine Learning*, Seoul, South Korea. PMLR 306, 2026. Copyright 2026 by the author(s).

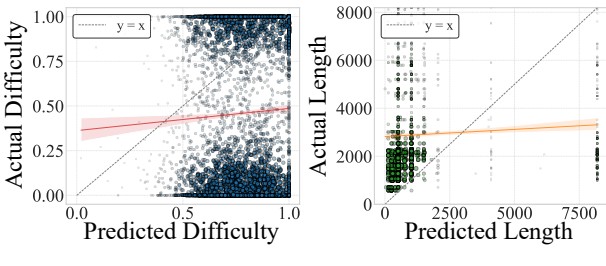

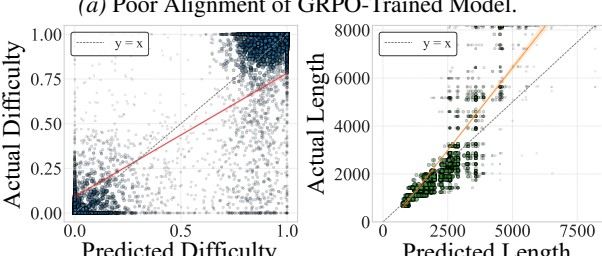

*(a)* Poor Alignment of GRPO-Trained Model.

*(b)* Enhanced Alignment of **MAPR**-Trained Model.

*Figure 1.* **Meta-Awareness of GRPO vs MAPR.** Predicted difficulty and solution length are elicited from both models using the same meta-prediction prompt, and the predictions are parsed from the model outputs. Difficulty is defined as the Pass@1 scores, while length refers to the model output token count. Note that jitter is applied to discrete difficulty values to aid density visualization.

large reasoning models (LRMs) (Guo et al., 2025a; Chen et al., 2025b) without an explicit critic model, have recently attracted considerable attention.

Beyond the success of LRMs, the paradigm of meta-awareness, which is the ability to recognize its own knowledge and ignorance, has drawn increasing attention from the research community (Sui et al., 2025; Ha et al., 2025; De Sabbata et al., 2024; Chen et al., 2025a; Liu et al., 2025c; Zhang et al., 2025a; Shen et al., 2025; Tu et al., 2025; Shi et al., 2025; Qu et al., 2025). However, existing approaches remain constrained by their reliance on external model, curated dataset, and reasoning pipelines that require human intervention.

To this end, we propose a novel RL framework, **M**eta-**A**wareness via **P**redictive **R**eward (**MAPR**), which formalizes meta-awareness in reasoning models by rewarding the internal consistency of self-generated signals, thereby eliminating the need for external supervision. Our method intro-

duces a self-predictive trajectory coupled with the primary reasoning path, enhancing the model's meta-awareness of its computational budget, knowledge boundaries, and cognitive strategy. These improved meta-predictions, shown in Figure 1, drive training efficiency through *predictive gating*, which prunes zero-variance prompts by identifying those that are either trivial or unsolvable, and *early cutoff*, which terminates long rollouts predicted to result in incorrect outcomes. Furthermore, the model can use the predicted notions to generate hints for the primary reasoning process.

Building on this foundation, we evaluate the effectiveness of our approach by combining it with GRPO and DAPO (Yu et al., 2025; Shao et al., 2024), showing that our method is not dependent on a specific policy gradient algorithm. Remarkably, **MAPR** achieves substantial improvements on mathematical benchmarks achieving the strongest performance among the methods compared under the same compute budget. Finally, predictive gating and early cutoff deliver significant efficiency gains, reaching the GRPO baseline performance 1.28× faster while achieving higher final accuracy.

The contributions of this paper can be summarized as follows:

- We introduce a predictive reward signal formulated as a parallel verification prompt, which enables the model to self-evaluate meta-awareness by alignment.

- We experimentally show that the meta-prediction directly drives performance gain through paired analysis.

- We propose **MAPR**-*efficient*, a post-training strategy with predictive gating and early cutoff, which achieves the strongest performance with minimal training compute.

## 2. Related Works

**Meta-Cognitive Learning** Meta-cognition is viewed as a prerequisite for self-improving LLMs (Liu & van der Schaar, 2025). Existing methods rely on extrinsic mechanisms with fixed action loops, limiting adaptability. Self-improving agents that plan, regulate, and reflect (Dong et al., 2025; Didolkar et al., 2025) or refine prompts via past reasoning (Qiu et al., 2025; Liu et al., 2025c) entangle control with reasoning, often causing interference. In contrast, our approach disentangles the meta and solution paths for stable training on meta-awareness.

Other works require curated datasets (Ha et al., 2025), or delegate control to external verifiers (Ma et al., 2025; He et al., 2025) or multi-agent systems (Wan et al., 2025; Yang & Thomason, 2025; Bilal et al., 2025; Khandelwal et al., 2025), reducing scalability of meta-cognitive training. Training-free heuristics such as confidence-based stopping (Yang et al., 2025b; Qiao et al., 2025; Lu et al., 2025) or

correctness checks (Ma et al., 2025) offer efficiency but lack genuine language-level meta-cognition. In contrast, our approach does not rely on human-curated reasoning pipelines, external verifiers, PRMs, or specialized datasets targeting meta-cognitive ability, but rather leverages the *self-generated signals to encourage alignment* between the meta-prediction and primary thinking process.

**Self-Control for Efficient Training** Another direction that leverages meta-cognition is to regulate reasoning efficiency by allocating budgets via difficulty assessment (Chen et al., 2025a; Tu et al., 2025; Shi et al., 2025; Qu et al., 2025; Huang et al., 2025; Ji et al., 2025; Di & JoyJiaoW, 2025; Han et al., 2024b; Fang et al., 2025; Yang et al., 2025c; Zhang et al., 2025b; Wang et al., 2025; Zhang et al., 2025a; Shen et al., 2025), constraining output length with penalties or fixed limits (Aggarwal & Welleck, 2025; Li et al., 2025; Xiang et al., 2025; Zhang & Zuo, 2025), and adaptively stopping, continuing, or reflecting for compact reasoning (Ha et al., 2025; Zhang et al., 2025c; Dai et al., 2025). While these methods improve inference-time efficiency, they focus on making reasoning shorter or faster at inference time, often at the cost of reduced reasoning performance. In contrast, we target *efficiency during the post-training phase*, achieving both efficiency and improved performance during model training rather than during inference.

## 3. MAPR: Meta-Awareness via Predictive Reward and MAPR-*efficient*

We first provide background on group relative policy optimization (GRPO) (Section 3.1). Then we show our method: (i) **MAPR**, which endows the LLM with the capability to perform accurate meta-predictions (Section 3.2); and (ii) **MAPR**-*efficient*, an efficiency-enhanced version that accelerates **MAPR** through predictive gating and early cutoff (Section 3.3).

### 3.1. Preliminaries

We present an overview of GRPO, which is a popular RL algorithm for post-training reasoning models. The old policy model $\pi_{\theta_{old}}$ produces a group of $G$ responses given prompt $\mathbf{q}$ from tasks $P(\mathcal{Q})$, creating rollouts $\mathcal{O} = \{\mathbf{o}_1, \cdots, \mathbf{o}_G\}$. Each response is assigned a reward $\{r_1, \cdots, r_G\}$ based on the rule-based verification of the extracted answer against the ground truth.

The objective of GRPO is formulated as,

$$\mathbb{E}_{\substack{\mathbf{q} \sim P(\mathcal{Q}) \\ \{\mathbf{o}_i\} \sim \pi_{\theta_{old}}(\cdot|\mathbf{q})}} \left[ \frac{1}{G} \sum_{i=1}^{G} \frac{1}{|\mathbf{o}_i|} \sum_{t=1}^{|\mathbf{o}_i|} \left( \mathcal{J}_{i,t}^{clip}(\theta) - \beta D_{KL}(\pi_\theta || \pi_{ref}) \right) \right] \quad (1)$$

where $\mathcal{J}_{i,t}^{clip}(\theta) = \min \left( \rho_{i,t}(\theta) \hat{A}_{i,t}, \text{clip}(\rho_{i,t}(\theta), 1-\epsilon, 1+\epsilon) \hat{A}_{i,t} \right)$.

Note that $\rho_{i,t}(\theta) = \frac{\pi_\theta(o_{i,t}|\mathbf{q}, \mathbf{o}_{i,<t})}{\pi_{\theta_{old}}(o_{i,t}|\mathbf{q}, \mathbf{o}_{i,<t})}$ denotes the importance

sampling ratio, and $\pi_\theta$ represents the current policy model. $\pi_{\text{ref}}$ is the reference model. $\text{clip}(\cdot)$ restricts the importance sampling ratio between $[1 - \epsilon, 1 + \epsilon]$. The advantage is calculated as $\hat{A}_{i,t} = \frac{r_i - \text{mean}(\{r_i\}_{i=1}^G)}{\text{std}(\{r_i\}_{i=1}^G)}$. Following the practice of recent GRPO variants (Liu et al., 2025a; Zhang & Zuo, 2025; Zheng et al., 2025; Yu et al., 2025), we set $\beta = 0$ to remove the KL divergence term.

### 3.2. MAPR: Designing Meta-Awareness via Predictive Reward

**Overall Pipeline of MAPR**  Building on the GRPO-based framework, the policy model is prompted with two distinct inputs: *solution* prompt $\mathbf{q}_{\text{sol}}$ and *meta* prompt $\mathbf{q}_{\text{meta}}$. The solution and meta rollouts are executed simultaneously, but the rewarding pipelines differ. Solution rollouts are verified against static ground truth using rule-based verification, while meta rollouts are verified against empirical statistics derived from the solution rollouts as dynamic ground truth.

The *solution* prompt $\mathbf{q}_{\text{sol}}$ instructs the model to solve the problem via chain-of-thought, generating a group of $G$ solution rollouts as detailed in Section 3.1. For meta-prediction verification, the average Pass@1 score over $G$ rollouts, $(p)$, and the range of output token lengths among correct rollouts, $([l_{\text{min}}, l_{\text{max}}])$, are extracted, and the entire responses, $(\mathcal{O})$, are saved for predicted notion verification.

Simultaneously, the *meta* prompt $\mathbf{q}_{\text{meta}}$ instructs the model to predict the expected difficulty as Pass@1 score $(\hat{p})$, the expected length of a correct response $(\hat{l})$, and a set of problem-solving notions $(\hat{\mathcal{G}}_{\text{notion}})$. We generate $M$ independent meta-rollouts,[1] and reward each based on how accurately it predicts the output length, problem difficulty, and used notions from the solution rollouts. The meta rewards, $\{r_1^{\text{meta}}, \ldots, r_M^{\text{meta}}\}$, are then normalized within the group of $M$ rollouts to compute advantages. The reward computation for each meta component is detailed below. For reproducibility, we provide the complete code snippet in Section C.

**Difficulty Reward.**  The difficulty alignment reward measures the proximity between the predicted pass-rate $\hat{p}$ and the actual pass-rate $p$. This is the proportion of correct answers among $G$ rollouts for question $q$. This allows the model to learn how hard the given question is for the current knowledge boundary of the model.

We compute the accuracy score as an exponential decay function of the normalized prediction error, given by

$$r_{\text{difficulty}} = 0.01^{|p-\hat{p}|}.$$

[1]The full meta-prediction prompt template is deferred to Section A.

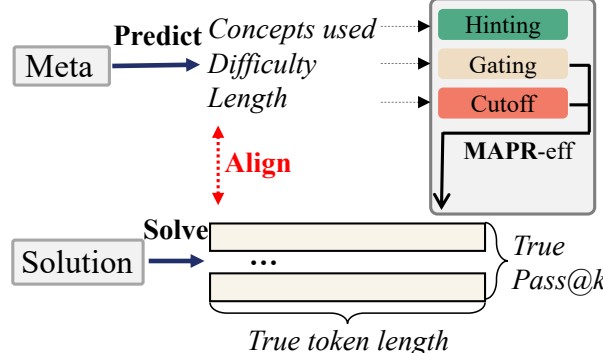

*Figure 2.* **Overall Framework of MAPR and MAPR-*efficient*** **MAPR** predicts and solves in parallel from given *meta* and *solution* prompts. The predicted values are verified against true pass@k, token length, and used concepts extracted from the solution rollouts. The efficient version, **MAPR-*efficient***, applies predictive gating and length cutoff for efficient training.

A deviation of a single unit in difficulty prediction approximately halves the reward with the base number 0.01, in order to strongly penalize higher errors in prediction.

**Length Reward.**  The length alignment reward checks whether the predicted length falls within the range of correct responses. Formally, we assign the reward if the predicted length $\hat{l}$ falls within the range of correct responses as

$$r_{\text{length}} = \mathbb{1}\big[l_{\text{min}} \leq \hat{l} \leq l_{\text{max}}\big].$$

If no correct solution exists, then we set the reward as 0.

**Notion Reward.**  The notion reward evaluates whether the predicted problem-solving notion emerges more frequently in correct rollouts than in incorrect ones. More formally, for a single notion $n \in \hat{\mathcal{G}}_{\text{notion}}$, we count the number of correct responses containing $n$ (denoted as $c_{\text{corr, n}}$) and the number of incorrect responses containing $n$ (denoted as $c_{\text{wrong, n}}$).

Then, the notion reward is defined as

$$r_{\text{notion}} = \mathbb{E}_{n \sim \hat{\mathcal{G}}_{\text{notion}}}\big[\mathbb{1}[c_{\text{corr, n}} > c_{\text{wrong, n}}]\big].$$

Notions present in the problem statement are excluded to prevent reward hacking, and lemma-based matching is used for counting.

Then, the meta reward is defined as the average of three components,

$$r^{\text{meta}} = \frac{r^{\text{length}} + r^{\text{difficulty}} + r^{\text{notion}}}{3}. \tag{2}$$

### 3.3. MAPR-*efficient*: Meta-based Active Control for Efficient Post-Training

**MAPR-*efficient*** is a variant of **MAPR** that can further boost training efficiency by leveraging the length and difficulty

*Table 1.* **Performance of GRPO and MAPR for Math benchmarks.** Pass@1 and Pass@8 scores are reported with standard deviations over 32 samplings. The overall performance of our method **MAPR** surpasses baseline GRPO method by a large margin.

| Benchmark | GRPO | | GRPO w/ MAPR | |
|---|---|---|---|---|
| | Pass@1 | Pass@8 | Pass@1 | Pass@8 |
| **Qwen3-4B Base Model** | | | | |
| AIME'24 | 17.50±4.00 | 33.60±5.96 | **26.15**±**3.32** (+ 49.43%) | **48.82**±**5.32** (+ 45.30%) |
| AIME'25 | 11.77±4.56 | 25.56±4.40 | **21.56**±**4.40** (+ 83.18%) | **37.17**±**3.63** (+ 45.42%) |
| AMC23 | 59.30±6.40 | 84.93±3.90 | **70.16**±**4.78** (+ 18.31%) | **93.18**±**1.90** (+ 9.71%) |
| MATH500 | 79.61±0.91 | 90.12±0.59 | **84.52**±**0.74** (+ 6.17%) | **93.74**±**0.42** (+ 4.02%) |
| Minerva | **42.27**±**1.53** | 59.70±0.91 | 41.12±2.00 (- 2.72%) | **63.78**±**1.35** (+ 6.83%) |
| Olympiad | 44.47±1.04 | 61.99±0.61 | **53.38**±**0.96** (+ 20.04%) | **69.74**±**0.69** (+ 12.50%) |
| **Average** | 42.49±3.07 | 59.31±2.73 | **49.48**±**2.70** (+ 16.45%) | **67.73**±**2.22** (+ 14.20%) |
| **Qwen3-8B Base Model** | | | | |
| AIME'24 | 28.54±4.12 | 53.96±4.07 | **34.17**±**5.54** (+ 19.72%) | **63.80**±**4.98** (+ 18.24%) |
| AIME'25 | 22.19±3.63 | 38.74±4.05 | **28.44**±**5.41** (+ 28.17%) | **45.96**±**4.41** (+ 18.64%) |
| AMC23 | 73.67±5.60 | 92.77±2.43 | **79.53**±**4.26** (+ 7.95%) | **94.39**±**1.80** (+ 1.75%) |
| MATH500 | 85.75±0.66 | 94.31±0.49 | **88.05**±**0.82** (+ 2.68%) | **95.35**±**0.49** (+ 1.1%) |
| Minerva | 43.21±2.12 | 64.00±1.14 | **47.21**±**1.74** (+ 9.26%) | **68.21**±**1.23** (+ 6.58%) |
| Olympiad | 54.03±1.22 | 70.04±0.70 | **56.86**±**0.85** (+ 5.24%) | **71.87**±**0.51** (+ 2.61%) |
| **Average** | 51.23±2.89 | 68.97±2.15 | **55.71**±**3.10** (+ 8.74%) | **73.26**±**2.24** (+ 6.22%) |
| **Qwen3-14B Base Model** | | | | |
| AIME'24 | 38.54±4.30 | 58.55±4.07 | **44.27**±**5.64** (+ 14.87%) | **68.30**±**3.57** (+ 16.65%) |
| AIME'25 | 27.92±4.69 | 45.56±3.87 | **31.25**±**5.12** (+ 11.93%) | **53.57**±**5.51** (+ 17.58%) |
| AMC23 | 81.56±4.98 | **96.20**±**1.66** | **86.02**±**4.16** (+ 5.47%) | 95.12±1.48 (- 1.12%) |
| MATH500 | 88.73±1.03 | 96.02±0.36 | **89.93**±**0.88** (+ 1.35%) | **96.39**±**0.34** (+ 0.38%) |
| Minerva | 45.03±1.73 | 66.42±1.06 | **50.36**±**1.53** (+ 11.84%) | **69.20**±**0.96** (+ 4.19%) |
| OlympiadMath | 59.04±0.90 | 73.03±0.58 | **61.59**±**0.89** (+ 4.32%) | **74.37**±**0.65** (+ 1.83%) |
| **Average** | 56.80±2.94 | 72.63±1.93 | **60.57**±**3.04** (+ 6.63%) | **76.15**±**2.09** (+ 4.85%) |

predictions.

**Overall Pipeline of MAPR-*efficient*** To encourage meta-awareness before accelerating the training phase, we first perform self-alignment based policy updates for the early $k$ steps of update with self-predictive alignment reward, until the policy model shows stable meta-prediction alignment with the true solution rollouts. After the $k$-th step, we switch to a non-parallel pipeline that executes meta-predictions first, for predictive gating, followed by solution rollouts, applying early length cutoff. We may also utilize the predicted notions to provide additional hints for the model in solving the questions.

**Predictive gating** acts as a pre-computation filter for tasks that are deemed either trivial or impossible to solve. For a given question $q$, gating engages only when the standard deviation across $M$ predicted pass-rates falls below $\sigma_{pg}$ and the average prediction is 0 or 1. Distinct from methods like

DAPO, which prune *after* expensive solution rollouts, our approach conserves computation by gating *before* the rollout phase. Since our primary objective is training efficiency, we employ static online gating; dynamically re-evaluating previously gated tasks would require periodic, costly meta-predictions on excluded data.

**Length cutoff** restricts generation to the predicted length, scaled by a margin $l_{LC}$. As the **MAPR** length reward incentivizes accurate prediction for correct rollouts, exceeding this threshold is highly unlikely to yield a correct answer, despite the cost of generating additional tokens. Therefore, **MAPR**-*efficient* utilizes the length prediction as a hard threshold to terminate rollouts once the limit is reached.

Additionally, **notion feed-in** is implemented by appending the hint "The problem could be solved using the following math notions" to the problem statement, providing auxiliary guidance during the solution rollout phase.

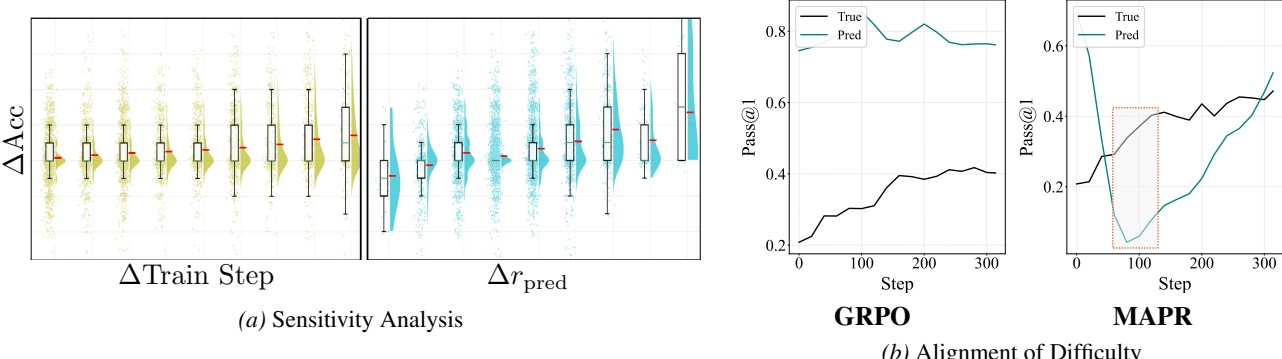

*(a)* Sensitivity Analysis

**GRPO**  **MAPR**

*(b)* Alignment of Difficulty

*Figure 3.* **Impact of Meta-Awareness on Training Dynamics.** (a) We observe a significantly steeper gradient for meta-awareness ($r_{\text{pred}}$) compared to training steps, suggesting that increased meta-awareness drives performance more effectively than training duration alone. (b) The **MAPR** Pass@1 surge (steps 80-120) coincides precisely with the *drop-then-align* phase in difficulty prediction (orange), implying that predictive calibration correlates strongly with performance increase.

## 4. Experiments

In this section, we provide the details of training and evaluation configuration in Section 4.1. Then we demonstrate the performance gain and efficiency driven by **MAPR** and **MAPR**-*efficient* in Section 4.2. In addition, we systematically analyze the components of our method through ablation studies in Section 4.3.

### 4.1. Training and Evaluation Details

**Training Details.** We use VeRL with the DeepScaleR (Luo et al., 2025) dataset, batch size 128, learning rate 1e-6, $10\%$ weight decay, maximum response length 8K, and GRPO without KL term. Training runs for one epoch (314 steps) using AdamW (Loshchilov & Hutter) with 20 warm-up steps, gradient clipping at 1.0, and clipping range for GRPO between $[\epsilon_{\text{low}} = 0.2, \epsilon_{\text{high}} = 0.28]$. The rollouts use temperature 1.0 and top-p value of 1.0. The number of rollouts is 16 for the response generation, and 8 for meta prediction.

**Evaluation Configuration.** We use the provided math scoring function in VeRL to measure the accuracy of the predicted answer and ground truth answer, sampling 32 responses, with 16k maximum response length and temperature set at 0.6.

We evaluate the performance of our method using six widely used mathematical reasoning benchmarks, AIME24, AIME25, AMC23, MATH500 (Hendrycks et al.), Minerva, and OlympiadBench (He et al., 2024). Experiments are conducted on Qwen3 8B base model unless otherwise stated.

### 4.2. Analysis of MAPR and MAPR-*efficient*

**MAPR Excels on Mathematical Benchmarks** **MAPR** outperforms the baseline in six math benchmarks - AIME24, AIME25, AMC23, MATH500, Minerva, and Olympiad-Bench (Table 1). Across all mathematical datasets, our

method **MAPR** shows great improvement over the baseline GRPO performance, showing an average improvement of 16.45% in Qwen3-4B model, 8.74% in Qwen3-8B model, and 6.63% in Qwen3-14B model. Among the six benchmarks, **MAPR** achieves the largest gains on intermediate to hard-level benchmarks (AIME, AMC, Olympiad, Minerva), while the performance boost for MATH500 shows performance saturation especially for large scaled model of 14B. We also demonstrate the superior ability of **MAPR** on out-of-domain benchmarks, across logical reasoning, scientific reasoning, and coding domains in Table 8.

**Meta-Awareness Directly Enhances Performance** In Figure 3a, we assess whether performance gains arise from improvements in the reward metric $r_{\text{pred}}$ or from extended training. We conduct a paired analysis comparing marginal accuracy gains ($\Delta$Acc) with both training steps and meta-awareness measured by $r_{\text{pred}}$. Checkpoints are sampled every 20 steps across six mathematical benchmarks, and for each question we pair model states $(t_1, t_2)$ from different steps to control for input variation.

We plot step differences versus accuracy gains using a jittered distribution. To isolate the effect of meta-awareness, we compute $\Delta r_{\text{pred}} = |r_{\text{pred}}^t - r_{\text{pred}}^q|$ and plot binned values against $\Delta$Acc. Accuracy improves more steeply with respect to $r_{\text{pred}}$ than with training steps, indicating that performance is more sensitive to meta-cognitive calibration than to additional training compute.

**Meta-prediction Dynamics During MAPR Training** As shown in Figure 3b, a critical divergence in training dynamics appears when analyzing the model's self-prediction of problem difficulty ($\hat{p}$) versus the true difficulty ($p$). GRPO exhibits consistent overconfidence, predicting a Pass@1 value exceeding 0.8, despite its true score remaining significantly lower.

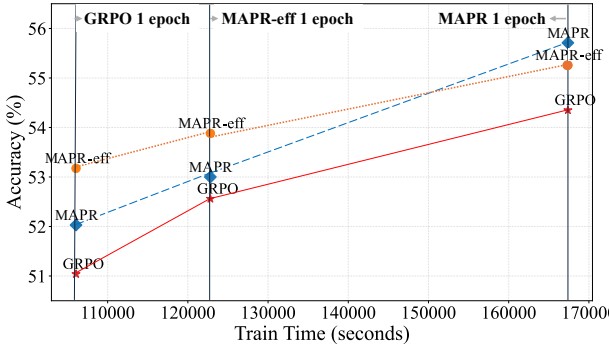

*Figure 4.* **Accuracy vs Wall Clock Time.** Average performance across six math benchmarks. Gray vertical lines indicate epoch milestones. Both **MAPR**-*efficient* and **MAPR** achieve Pareto-superiority over the GRPO baseline, showing higher accuracy for the same compute expenditure.

| Metric | Value |
|---|---|
| Precision | 0.9417 |
| Recall | 0.8739 |
| F1 Score | 0.9065 |

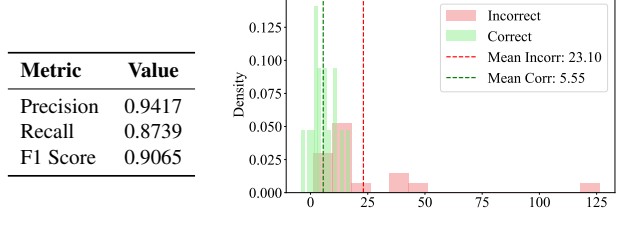

(a) Predictive Gating  (b) Length Cutoff

*Figure 5.* (a) Accuracy of Predictive Gating (b) Standard Error of Length Cutoff from **MAPR**-*efficient*.

In contrast, while **MAPR** also begins with initial overconfidence, the meta-awareness objective drives a corrective *drop-then-align* behavior in Figure 3b. The predicted difficulty drops sharply until step 80, recalibrating to match the true Pass@1. Crucially, this coincides with the rapid ascent in the true Pass@1 score, suggesting that accurate self-assessment correlates with the performance gains observed in **MAPR**. A similar tendency is observed in length prediction ($l$ vs $\hat{l}$), as shown in Section D.

**MAPR-*efficient* Achieves the Strongest Performance with Minimal Training Compute**    Under the same compute budget (wall clock time), we demonstrate that **MAPR** and **MAPR**-*efficient* surpass the performance of GRPO. Figure 4 reports the average accuracy across six mathematical benchmarks at wall-clock times corresponding to one epoch of GRPO, **MAPR**-*efficient*, and **MAPR**. At each matched compute budget, **MAPR**-*efficient* and **MAPR** outperform GRPO by a substantial margin. This proves the efficacy of our method in achieving a large performance gain even under same amount of training compute time.

**Prediction Performance of PG and LC**    To evaluate the reliability of predictive gating and length cutoff in **MAPR**-*efficient*, we compare gating and cutoff decisions against the ground-truth. Using an unseen portion of the DeepScaleR training dataset, Figure 5a reports the performance of predictive gating in terms of precision, recall, and F1 score against true zero-variance questions. These metrics evaluate whether the predicted difficulty value of 0 or 1 with a standard deviation below $\sigma_{\text{pg}}$ matches the true zero variance. Moreover, Figure 5b shows the standardized error value of the length cutoff decision. The standard error is calculated as $\left( \mathbb{E}_{i \sim \mathcal{O}_{\text{sol}}}(l_i) - \mathbb{E}_{i \sim \mathcal{O}_{\text{meta}}}(\hat{l}_i) \right) / \sigma(\hat{l})$, which quantifies the deviation of the true length from the prediction, normalized by meta-prediction uncertainty. The distribution shows the

standard error from correct rollouts (green) and incorrect rollouts (red). While the distribution for correct rollouts is centered around zero error (0), the distribution for incorrect rollouts is shifted toward larger values. This demonstrates that the length predictions are highly accurate, and the cutoff strategy effectively prevents the model from generating futile extra tokens that would lead to wrong answers. The distribution of standard error values toward the positive range indicates that the length cutoff mechanism serves as a highly effective means of conserving tokens.

### 4.3. Ablation Studies

**Meta-prediction Components**    To attribute performance improvements to individual factors, we employ a Shapley-$R^2$ decomposition based on linear regression and plot the value in Figure 6a. We define the design matrix as the feature matrix composed of paired differences ($\Delta r_{\text{difficulty}}, \Delta r_{\text{length}}, \Delta r_{\text{notion}}, \Delta \text{step}$), and let $\Delta \text{Acc}$ be the target variable to compute the Shapley-$R^2$ values. The details are deferred due to space constraints.

Moreover, we conduct an ablation study by training **MAPR** with each of the three components and using all three. The results in Figure 6b show that using all three components of meta-prediction shows overall superior performance over all benchmarks.

**Number of Meta Rollouts**    In Figure 6c, we analyze the effect of reducing the number of meta-prediction paths from 16, which is the default rollout number for the primary solution path. Evaluation shows that using 8 rollouts for meta-prediction, in combination with 16 rollouts for a solution path, yields the optimal result in terms of both training compute and performance.

Introducing the meta-prediction path requires the policy model to generate additional meta-predictions on the solution length, pass-rate, and high-level concepts. However, we show that the average token length and number of rollouts additionally required for meta-predictions only amount to 15.5% of total rollout compute as shown in Table 2.

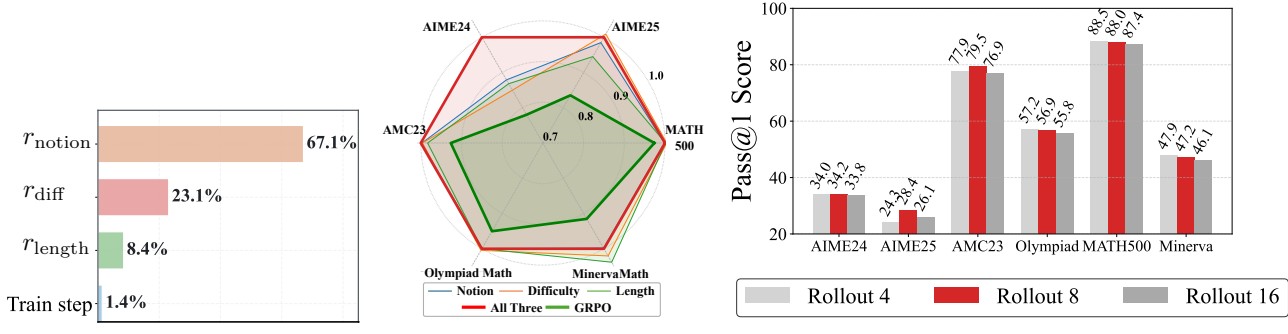

*(a)* Shapley $R^2$ Analysis on component-wise contributions.

*(b)* Ablation of meta components (Maximum set to the score of 'All three' components for visualization purpose).

*(c)* Ablation on number of meta-prediction rollouts.

*Figure 6.* **Component Analysis and Ablation Studies** The contribution of our meta-aware predictive reward components are analyzed using Shapley values (left), meta-component ablations (center), and different numbers of meta-prediction rollouts (right).

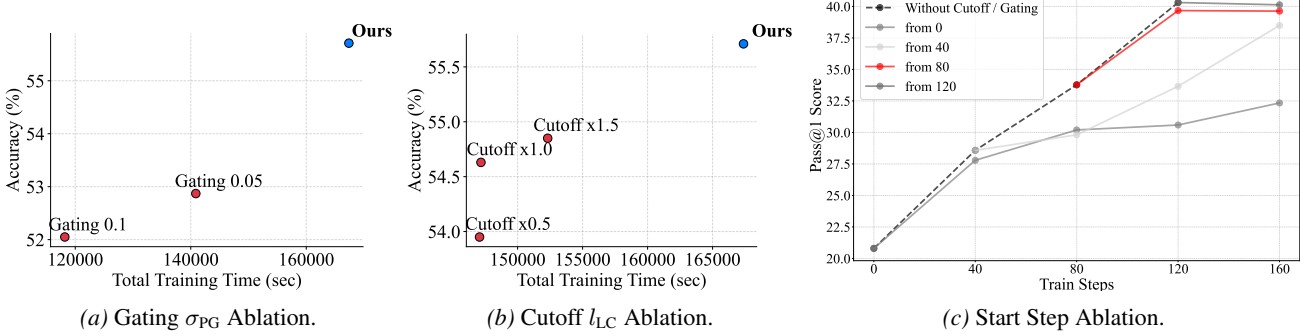

*(a)* Gating $\sigma_{\mathrm{PG}}$ Ablation.

*(b)* Cutoff $l_{\mathrm{LC}}$ Ablation.

*(c)* Start Step Ablation.

*Figure 7.* **Ablation on Hyper-parameters for MAPR-*efficient*.** Pass@1 scores over choices of (a) predictive gating (PG), (b) length cutoff (LC) and (c) start step (k).

*Table 2.* Comparison on token length between original rollouts and meta prediction rollouts. The additional tokens required for meta-prediction account for only 15.5% of the total rollout tokens.

| | Avg. Tokens | Rollout No. | Proportion |
|---|---|---|---|
| Solution | 6251 | 16 | 84.5% |
| Meta-Pred | 2293 | 8 | 15.5% |

**Hyper-parameters for MAPR-*efficient* ($\sigma_{\mathrm{PG}}$, $l_{\mathrm{LC}}$)** Following an initial $k$-step training phase for length and difficulty meta-prediction, **MAPR-*efficient*** applies predictive gating ($\sigma_{\mathrm{PG}}$) and length cutoff ($l_{\mathrm{LC}}$).

In Figure 7a, we evaluate the impact of the predictive gating parameter, $\sigma_{\mathrm{PG}}$, which determines the threshold for skipping prompts based on the standard deviation of predicted difficulty. Lower values of $\sigma_{\mathrm{PG}}$ ensure that gating only occurs when the meta-predictions have low variance regarding task difficulty. Similarly, Figure 7b illustrates the effect of the length cutoff margin, $l_{\mathrm{LC}}$, a multiplier that scales the threshold for early trace termination. While both PG and LC incur a marginal performance trade-off relative to **MAPR**, these costs are effectively offset by substantial

gains in training efficiency and reduced wall-clock time.

**Optimizing the Transition Step $k$** The timing of these efficiency mechanisms is critical, as **MAPR-*efficient*** requires an initial calibration period. Figure 7c compares the performance of four start-step variants ($k \in \{0, 40, 80, 120\}$) across 160 training steps. We observe that while premature activation ($k < 80$) slightly degrades final accuracy, initiating predictive gating and length cutoff at step 80 achieves performance parity with later starts (e.g., $k = 120$) while providing earlier computational savings. We set $k = 80$ for **MAPR-*efficient***, as it represents the optimal balance between meta-prediction calibration and resource efficiency. We show that this result is consistent across model sizes in Table 3. All configurations converge to consistent final performance even when starting from different start steps for 14B model.

**Base Number** The choice of base number for difficulty reward $r_{\mathrm{difficulty}}$ is set as 0.01 to halve the reward per unit difference between predicted and true difficulty. We test the robustness of our method **MAPR** with different base

numbers 0.005, 0.01, and 0.02 on 4B and 8B scale models. As shown in Table 4, the results with different base numbers show consistent scores across model sizes and base numbers, except for the extreme value of 0.005 on the 4B model, which degrades performance.

**Meta-Predicted Notion Feed-in** We test whether the notions generated from meta-predictions serve as an auxiliary hint for the original solution rollouts by incorporating them into the question using prompt format: Question + 'The problem could be solved using following math notions'. To examine the impact of such notion feed-in on performance, we conduct the following experiment which uses the predicted notions as hints to the problem-solving phase.

As shown in Table 5, incorporating notion feed-in (**MAPR** + NF) produces mixed results with nearly unchanged average performance compared to the variant without notion feed-in (**MAPR**). This suggests a high degree of information overlap, implying that the model likely implicitly possesses these concepts through **MAPR**, which enhances meta-awareness, making explicit hinting redundant. Therefore, we test whether the extracted notions boost the performance of a baseline GRPO model. The notions are extracted from the MAPR model and fed into a separately trained GRPO model using keywords with high notion reward scores. Although this setting is far from practical deployment, as it requires cross-model notion extraction and transfer, the substantial improvement achieved by GRPO + NF demonstrates that the extracted notions are highly effective in enhancing reasoning performance.

**Ablation on RL Algorithm** **MAPR** is flexibly applicable to GRPO variants. We show the superiority of **MAPR** combined with the DAPO algorithm in Table 6. Unlike DAPO, which requires a redundant sampling phase to filter out tasks with zero-variance, our method is able to bypass the sampling for solution rollouts and preemptively gate such tasks. Even with greater efficiency, **MAPR** outperforms DAPO on all six mathematical benchmarks by a large margin.

We train Qwen3-8B-Base with DAPO for three epochs (315 steps), which is equivalent to one epoch of GRPO (314 steps) in terms of the total number of gradient updates. Moreover, we disable the overlong reward shaping term in DAPO. In our setting, this term imposes an overly strong length constraint, which prevents the model from sufficiently increasing its reasoning depth. Empirically, we observe that

*Table 3.* Transitioning step ablation on Qwen3-14B Base Model.

| Start Step | AIME24 | AIME25 | AMC23 | Avg |
|---|---|---|---|---|
| 0 | 35.83 | 26.04 | 75.78 | 45.88 |
| 40 | 33.12 | 25.83 | 75.31 | 44.75 |
| 80 | 34.27 | 26.98 | 75.94 | 45.73 |
| 120 | 30.52 | 27.40 | 77.34 | 45.09 |

*Table 4.* Performance comparison across different base numbers for 4B and 8B models.

*(a)* **Qwen3-4B Base Model**

| Base Num | AIME'24 | AIME'25 | AMC'23 | MATH500 | Minerva | Olympiad | Avg |
|---|---|---|---|---|---|---|---|
| 0.005 | 16.98 | 14.58 | 61.95 | 79.62 | **43.55** | 45.21 | 43.65 |
| 0.01 | 26.15 | 21.56 | 70.16 | 84.52 | 41.12 | **53.38** | 49.48 |
| 0.02 | **26.77** | **23.33** | **70.47** | **84.84** | 43.11 | 53.24 | **50.29** |

*(b)* **Qwen3-8B Base Model**

| Base Num | AIME'24 | AIME'25 | AMC'23 | MATH500 | Minerva | Olympiad | Avg |
|---|---|---|---|---|---|---|---|
| 0.005 | **34.38** | 25.73 | 78.05 | **88.34** | **48.35** | **57.61** | 55.41 |
| 0.01 | 34.17 | **28.44** | 79.53 | 88.05 | 47.21 | 56.86 | **55.71** |
| 0.02 | 33.54 | 24.17 | **79.92** | 87.74 | 46.19 | 56.51 | 54.68 |

*Table 5.* **Performance of MAPR on Qwen3-8B across six mathematical benchmarks.** All metrics are **Pass@1**. NF denotes Notion-FeedIn.

| | GRPO | GRPO + NF | MAPR | MAPR + NF |
|---|---|---|---|---|
| AIME'24 | 28.54±4.12 | 33.96±5.88 | 34.17±5.54 | **35.10±4.96** |
| AIME'25 | 22.19±3.63 | 23.85±3.64 | **28.44±5.41** | 25.94±4.34 |
| AMC'23 | 73.67±5.60 | 77.97±5.08 | **79.53±4.26** | 78.91±5.01 |
| MATH500 | 85.75±0.66 | 86.52±1.14 | 88.05±0.82 | **88.51±0.79** |
| Minerva | 43.21±2.12 | 45.44±1.54 | 47.21±1.74 | **48.38±1.33** |
| Olympiad | 54.03±1.22 | 56.63±1.10 | 56.86±0.85 | **57.06±0.97** |

keeping this term results in lower final performance. We therefore remove it to avoid unnecessarily restricting the model's reasoning capacity under our training configuration.

**Ablation across Different Model Families** Our method also demonstrates consistent improvements when applied to different model families, Llama 3.1 8B Instruct (Grattafiori et al., 2024) and Gemma 2 9B IT (Team et al., 2024). Unlike Qwen3 models, which are explicitly trained on long CoT reasoning dataset, these two model families are relatively under-trained on mathematical reasoning dataset. Therefore, following the convention of existing works (Zhu et al., 2025; Liu et al., 2025b), we train both models on easier dataset, train split of MATH dataset, for three epochs. All the other configurations are kept the same. In Table 7, we report the evaluation results on AMC'23, MATH500, Minerva, and OlympiadBench, excluding AIME'24 and AIME'25 because the accuracy of both methods remains near zero. Overall, our method achieves consistent gains not only for Qwen3 but also for Llama 3.1 and Gemma 2 models.

## Conclusion

We present **MAPR**, a meta-aware reinforcement learning framework that fosters meta-cognitive ability by self-alignment. By incorporating information obtained by meta-thinking trajectories into training, our method enables stable and efficient optimization by integrating predictive gating and early cutoff. Empirically, **MAPR** accelerates RL-based post-training while improving both in-domain and out-of-domain performance, demonstrating notable gains in accu-

*Table 6.* Performance comparison of **MAPR** with DAPO, trained with Qwen3-8B base model.

| | DAPO | | DAPO + MAPR | |
|---|---|---|---|---|
| **Benchmark** | Pass@1 | Pass@8 | Pass@1 | Pass@8 |
| AIME'24 | $29.48_{\pm4.04}$ | $52.54_{\pm3.99}$ | $\mathbf{36.56}_{\pm\mathbf{5.97}}$ (+ 24.02%) | $\mathbf{66.28}_{\pm\mathbf{3.67}}$ (+ 26.15%) |
| AIME'25 | $23.75_{\pm3.83}$ | $37.00_{\pm2.96}$ | $\mathbf{25.94}_{\pm\mathbf{4.39}}$ (+ 9.22%) | $\mathbf{42.04}_{\pm\mathbf{2.83}}$ (+ 13.62%) |
| AMC'23 | $78.12_{\pm5.06}$ | $94.86_{\pm1.98}$ | $\mathbf{78.52}_{\pm\mathbf{4.56}}$ (+ 0.51%) | $\mathbf{95.14}_{\pm\mathbf{1.97}}$ (+ 0.29%) |
| MATH500 | $87.44_{\pm0.74}$ | $94.40_{\pm0.38}$ | $\mathbf{88.96}_{\pm\mathbf{0.85}}$ (+ 1.74%) | $\mathbf{95.10}_{\pm\mathbf{0.39}}$ (+ 0.74%) |
| Minerva | $45.22_{\pm1.80}$ | $65.43_{\pm1.04}$ | $\mathbf{47.97}_{\pm\mathbf{2.10}}$ (+ 6.08%) | $\mathbf{68.77}_{\pm\mathbf{0.99}}$ (+ 5.10%) |
| Olympiad | $55.97_{\pm0.89}$ | $71.13_{\pm0.63}$ | $\mathbf{57.53}_{\pm\mathbf{1.08}}$ (+ 2.79%) | $\mathbf{73.61}_{\pm\mathbf{0.73}}$ (+ 3.49%) |
| **Average** | $53.33_{\pm2.73}$ | $69.23_{\pm1.83}$ | $\mathbf{55.01}_{\pm\mathbf{3.16}}$ (+ 3.15%) | $\mathbf{73.49}_{\pm\mathbf{1.76}}$ (+ 6.15%) |

*Table 7.* Comparative performance of **MAPR** and GRPO across model variants.

*(a)* Llama 3.1 8B Instruct (3 Epochs / 174 steps)

| | GRPO | | MAPR | |
|---|---|---|---|---|
| **Benchmark** | Pass@1 | Pass@8 | Pass@1 | Pass@8 |
| AMC'23 | $25.23_{\pm4.16}$ | $42.92_{\pm3.23}$ | $\mathbf{31.02}_{\pm\mathbf{3.88}}$ | $\mathbf{56.52}_{\pm\mathbf{3.42}}$ |
| Math500 | $52.80_{\pm1.47}$ | $71.27_{\pm0.91}$ | $\mathbf{53.54}_{\pm\mathbf{1.14}}$ | $\mathbf{71.68}_{\pm\mathbf{0.79}}$ |
| Minerva | $31.70_{\pm1.69}$ | $50.15_{\pm1.21}$ | $\mathbf{31.86}_{\pm\mathbf{1.54}}$ | $\mathbf{50.39}_{\pm\mathbf{1.19}}$ |
| Olympiad | $19.39_{\pm0.87}$ | $32.36_{\pm0.66}$ | $\mathbf{19.89}_{\pm\mathbf{0.79}}$ | $\mathbf{35.61}_{\pm\mathbf{0.70}}$ |

*(b)* Gemma 2 9B IT (3 Epochs / 174 steps)

| | GRPO | | MAPR | |
|---|---|---|---|---|
| **Benchmark** | Pass@1 | Pass@8 | Pass@1 | Pass@8 |
| AMC'23 | $26.88_{\pm4.80}$ | $46.48_{\pm3.84}$ | $\mathbf{29.22}_{\pm\mathbf{4.52}}$ | $\mathbf{58.88}_{\pm\mathbf{3.54}}$ |
| Math500 | $54.07_{\pm0.90}$ | $71.70_{\pm0.79}$ | $\mathbf{57.24}_{\pm\mathbf{1.16}}$ | $\mathbf{76.57}_{\pm\mathbf{0.75}}$ |
| Minerva | $\mathbf{34.09}_{\pm\mathbf{1.54}}$ | $47.90_{\pm0.98}$ | $33.58_{\pm1.85}$ | $\mathbf{49.57}_{\pm\mathbf{1.11}}$ |
| Olympiad | $20.29_{\pm0.95}$ | $36.56_{\pm0.75}$ | $\mathbf{21.66}_{\pm\mathbf{0.80}}$ | $\mathbf{38.59}_{\pm\mathbf{0.76}}$ |

racy and generalization. These results highlight the promise of meta-prediction as a principled avenue for enhancing reasoning models.

## Impact Statement

This paper presents **MAPR**, a framework designed to verify and utilize meta-awareness in reasoning models. The primary broader impact of our work lies in improving computational efficiency for large language models. By enabling models to self-regulate, such as determining optimal thinking duration and filtering out unsolvable prompts, our approach significantly reduces the computational resources required for both training and inference. This contributes to reducing the environmental footprint associated with developing and deploying large-scale reasoning systems. While advancing reasoning capabilities generally implies the need for careful consideration of dual-use risks, our work specifically focuses on internal verification and efficiency, and we do not foresee specific negative societal consequences unique to this method.

## Acknowledgement

This work was supported by Institute for Information & communications Technology Planning & Evaluation (IITP) grant funded by the Korea government (MSIT) (RS-2019-II190075, Artificial Intelligence Graduate School Program (KAIST)) and National Research Foundation of Korea (NRF) grant (No.RS-2023-00209060, A Study on Optimization and Network Interpretation Method for Large-Scale Machine Learning) funded by the Korea government (MSIT).

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

# A. Default Meta-prediction Prompt for MAPR

---

**Prompt**

**[System]:**
You are a helpful assistant.
**[User]:**
Think step-by-step between <meta> and </meta>, ensuring comprehensive and detailed reasoning especially for determining the pass_rate and solution_length values. For each component (math_notion, pass_rate, solution_length), provide a comprehensive illustration or example during your reasoning in the <meta> section to clarify how each value is decided. When determining math_notion, ensure that the notions listed do not directly include the notions already written in the problem statement. After </meta>, return a JSON object with three keys:
- math_notion (list[str])
- pass_rate (integer from 0 to 8)
- solution_length (integer from 128 to {max_response_length})

Problem: {problem}

---

In the meta-prediction prompt, `math_notion` is predicted as a `list[str]`, where each element denotes a mathematical notion required to solve the problem. We avoid predicting a continuous value (or an overly fine-grained scale) for `pass_rate`, since it can introduce unnecessary variance and instability in the predicted difficulty. Instead, the prompt restricts `pass_rate` to an integer in $\{0,\ldots,8\}$. When computing the reward, we normalize this value by dividing it by 8. Finally, `solution_length` is predicted as an integer between 128 and the maximum response length of the corresponding training setup.

# B. Out-of-Domain Evaluation

*Table 8.* **Performance of GRPO and MAPR in Out-of-Domain benchmarks.** Results are reported as pass@1 score.

| Logical Reasoning | | | Scientific Reasoning | | | Coding | | |
|---|---|---|---|---|---|---|---|---|
| **Benchmark** | **GRPO** | **w/ MAPR** | **Benchmark** | **GRPO** | **w/ MAPR** | **Benchmark** | **GRPO** | **w/ MAPR** |
| ProntoQA | 90.56 | **93.74** | GPQA Diamond | 51.72 | **53.72** | EvalPlus | 77.32 | **77.66** |
| ProofWriter | 72.27 | **73.23** | R-Bench | 60.69 | **61.68** | CRUX-O | 72.72 | **73.39** |
| FOLIO | 69.16 | **69.24** | ARC-Challenge | 93.10 | **93.13** | MBPP | 71.84 | **72.97** |
| Logi. Deduct | 80.81 | **81.03** | SciBench | 28.33 | **29.64** | LiveCodeBench | 31.49 | **31.61** |
| AR-LSAT | 37.00 | **38.00** | | | | | | |
| **Avg.** | 69.96 | **71.05** | **Avg.** | 58.46 | **59.54** | **Avg.** | 63.34 | **63.91** |

**MAPR Performance in Out-of-Domain Benchmarks**    Meta-awareness also improves the reasoning model's generalization on out-of-domain logical reasoning, scientific reasoning, and coding benchmarks as shown in Table 8. For logical reasoning domain, we follow the setup of (Pan et al., 2023) and test on ProntoQA (Saparov & He), ProofWriter (Tafjord et al., 2021), FOLIO (Han et al., 2024a), LogicalDeduction (Srivastava et al.), and AR-LSAT (Zhong et al., 2022). For scientific reasoning, we use GPQA Diamond (Rein et al., 2024), R-Bench (Guo et al., 2025b), ARC-Challenge (Clark et al., 2018), and SciBench (Wang et al., 2024). For coding, we evaluate on EvalPlus (Liu et al., 2023), CRUX-O (Gu et al., 2024), MBPP (Austin et al., 2021), and LiveCodeBench (Jain et al., 2025). Although **MAPR** is not explicitly trained for generalization, strengthening meta-awareness consistently enhances out-of-domain performance. The base model is Qwen3-14B-Base, with the same training and evaluation configurations stated in the experiments section.

# C. Meta Reward Code Snippet

The implementation of our scoring mechanism is shown in the snippet below. We calculate a composite score based on the presence of mathematical notions, the length of the solution, and the difficulty pass rate.

```python
def compute_score(solution_str: dict, ground_truth: dict) -> float:

    # --- 1. Check Input Availability ---
    has_notion = "math_notion" in solution_str
    has_length = isinstance(solution_str.get("solution_length"), int)
    has_diff = isinstance(solution_str.get("pass_rate"), int)

    notion_score, length_score, acc_score = 0, 0, 0

    # --- 2. Calculate Notion Score ---
    if has_notion:
        pred_notions = solution_str["math_notion"]
        # Normalize to list
        if isinstance(pred_notions, str):
            pred_notions = [n.strip("[] ") for n in pred_notions.split(",")]

        # Filter notions already in problem text
        pred_notions = [n for n in pred_notions
                    if n not in ground_truth["problem"]]

        # Count occurrences (Pos for correct resp, Neg for incorrect)
        notion_counts = {n: 0 for n in pred_notions}
        for resp, correct in zip(ground_truth["response"], ground_truth["score"]):
            for n in pred_notions:
                if n in resp:
                    notion_counts[n] += (1 if correct == 1 else -1)

        # Score is ratio of notions with positive net utility
        scores = [1 if cnt > 0 else 0 for cnt in notion_counts.values()]
        if scores:
            notion_score = sum(scores) / len(scores)

    # --- 3. Calculate Length Score ---
    if has_length:
        correct_lens = [l for l, s in zip(ground_truth["length"],
                    ground_truth["score"]) if s == 1]
        p_len = solution_str['solution_length']
        if correct_lens:
            # Check if predicted length is within range of correct answers
            min_l, max_l = min(correct_lens), max(correct_lens)
            length_score = int(min_l < p_len < max_l)

    # --- 4. Calculate Accuracy Score ---
    if has_diff:
        avg_score = sum(ground_truth["score"]) / len(ground_truth["score"])
        pred_score = solution_str["pass_rate"] / 8.0
        acc_score = (0.01) ** abs(avg_score - pred_score)

    return (notion_score + acc_score + length_score) / 3
```

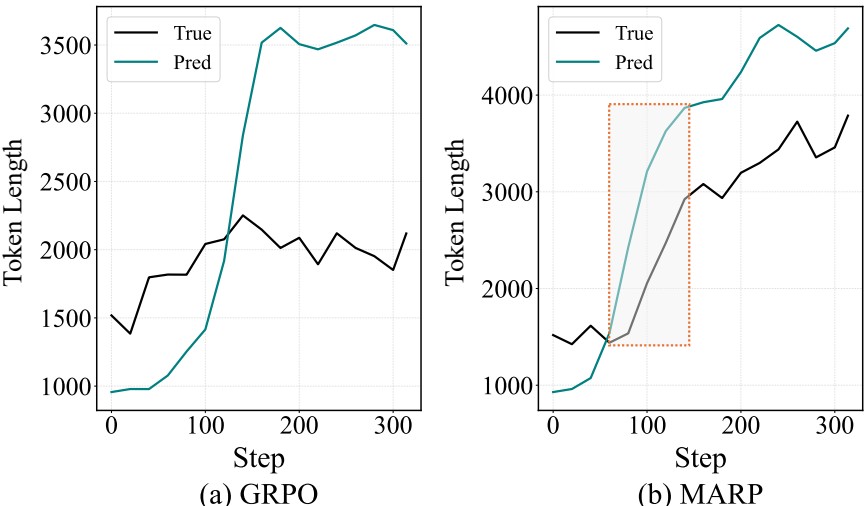

*Figure 8.* **Alignment of Length**. The MAPR solution length surge coincides with the rise-and-align phase of the predicted solution length, which corrects its initial underestimation and approaches the actual length, implying that predictive calibration correlates strongly with performance gains.

## D. Length Prediction and Training Dynamics

Similar to the observations on the difficulty prediction and training dynamics, we observe a surge in predicted length from the initial incorrect and underestimated state coincides with the rapid gain in performance during the training phase. This observation, coupled with the similar tendency in difficulty estimation, implies that calibration in the model's meta-awareness induces performance gain in reinforcement learning.

# E. Shapley $R^2$ Computation Details

We first fit a linear model using all $p$ features to obtain the full-model coefficient of determination $R^2_{\text{full}}$. To compute feature-level contributions, we consider all permutations of the feature set. For each permutation, features are added sequentially to the model, and the marginal increase in $R^2$ upon adding feature $j$ is recorded. The Shapley contribution of feature $j$ is then defined as the average of its marginal $R^2$ gains over all permutations. This decomposition yields an additive attribution of $R^2_{\text{full}}$, providing a principled measure of each factor's explanatory power while accounting for feature interactions and ordering effects.

# F. Discussions

*Table 9.* Examples of cross-domain notion prediction on coding and science tasks.

| Domain | Task | Extracted Notions |
|---|---|---|
| Coding | `maximum-strength-of-a-group` `find-the-longest-equal-subarray` `greatest-common-divisor-traversal` | array manipulation, mathematical operations, dynamic programming, greedy algorithms sliding window, hash map, two pointers graph traversal, GCD calculation, prime factorization |
| Science | Quantum mechanics problem Organic chemistry synthesis Gene interaction problem | quantum mechanics, Heisenberg uncertainty principle organic chemistry, Grignard reactions, oxidation, reaction mechanisms epistasis, transcription factor, gene redundancy |

Our notion-based reward formulation is not restricted to mathematical reasoning and can naturally generalize to other domains such as coding and science question answering. In coding tasks, notions correspond to high-level algorithmic concepts and data structure patterns, while in scientific reasoning they map to domain-specific scientific principles and terminology. To verify this, we apply a simple prompt adaptation without additional domain-specific training and analyze the generated notion predictions across domains. As shown in Table 9, the model consistently extracts meaningful and task-relevant notions for both coding and science problems, suggesting that notion prediction captures transferable high-level semantic abstractions beyond mathematics.

