# OpenReview forum: "Verifying Meta-Awareness via Predictive Rewards in Reasoning Models"
_ICML.cc/2026/Conference — ICML 2026 regular_

### Official Review · Reviewer_JAqg · 2026-03-07

**Soundness:** 4
**Presentation:** 2
**Significance:** 4
**Originality:** 4
**Overall Recommendation:** 5
**Confidence:** 3

**Summary:**

In training, the fundamental idea in the paper is to run two tasks in parallel: one, as usual, to answer a prompt (here to solve a problem); the other is to predict the difficulty of the problem, the length needed to solve it and the notions that will be needed in order to solve the problem. In training the loss fct rewards success in solving the problem but also if the predictions were accurate enough. In inference, the predictions are actively used to control the solving of the problem: if the solution is predicted to be very difficult then this triggers the solver to check more alternative paths or, if it is predicted to be easy, it indicates to reduce unnecessary steps. The prediction that certain notions should be present in the solution could also be used as guidance for the solver (but not in the paper). Experiments on math benchmarks show impressive performance improvements.

**Compliance With Llm Reviewing Policy:**

Affirmed.

**Final Justification:**

Authors answer very well my points, incl clearer distinction of MAPR and MAPR-efficient.

**Key Questions For Authors:**

1. I would like to see the full loss function, so the extension of (1) to MAPR, to understand precisely the training.
2. I would like to see the precise algorithm in inference, the interaction of the two components.
3. There are many ways to merge the three rewrds, your (2) is one. Why this one? Have you tried other weights? This might also influence your analysis in 4.3.
4. I think you can use you predictions in many other ways. Would you like to discuss this plrease?
5. You could also predict other things. Could you give examples?
6. I would like to see a more evidence that your predicitons (length and difficulty) are good enough. And when are they not good enough?
7. How does your method depend on the quality of the solver? I imagine that if the solver is bad, predictions are not going to help much. If this intuition is true, what does it mean for your method?

**Limitations:**

Only math tasks.

**Strengths And Weaknesses:**

Excellent original idea: on-the-go prediction of difficulty, length needed and notions required. The paper shows that it is possible to make such predictions well enough in useful time, and that it is possible to use these predicitons in inference to improve performance and efficiency. Even if the experiments are concentrated on math tasks, the evidence of advantage is clear. Idea has more potential too.
The presentation should be improved. Te way I understand this is that MAPR is used in training, while MAPR-effficient is the version used in inference. If I am wrong, the presentation is not clear enough; if I am right, then I would suggest that you write this more clearly. It is not just a way to improve efficiency, it is more! Your idea opens many additional opportunities, as you can use your predictions in many more ways.
Regarding significance, I am very supportive of your idea, if you can show that predictions are good, even with little data in inference.

---

> ### Author Rebuttal · Authors · 2026-03-31
>
> Thank you for the suggestion to clearly distinguish MAPR and MAPR-efficient. Both are training time methods, where MAPR use parallel solution and meta-prediction paths to improve model performance. MAPR-efficient is a efficient version of MAPR that additionally applies predictive gating and length cutoff during rollout phase to accelerate the training phase. We will clarify this more explicitly in the revision.
>
> ---
> ### **Point 1: Full Loss Function of MAPR**
> Thank you for the helpful suggestion. The full loss formulation can be written as follows:
> $$
> L_{MAPR}(\theta) = L_{sol}(\theta) + L_{meta}(\theta)
> $$
> The solution-generation objective is defined as
> $$L_{sol}(\theta) =E [(1/G) \sum_{i=1}^G (1/|o_i|) \sum_{t=1}^{|o_i|}( J_{sol,i,t}(\theta) - \beta D_{KL}(\pi_\theta || \pi_ref) )]$$
> where
> $$J_{sol,i,t}(\theta) =\mathrm{min}(\rho_{i,t}(\theta) A_{sol,i,t},\mathrm{clip}(\rho_{i,t}(\theta), 1-\epsilon_{low}, 1+\epsilon_{high}) A_{sol,i,t})$$
> Similarly, the meta-prediction objective is given by
> $$L_{meta}(\theta) =E [(1/M) \sum_{j=1}^M (1/|m_j|) \sum_{t=1}^{|m_j|}( J_{meta,j,t}(\theta) - \beta D_{KL}(\pi_\theta || \pi_ref) )]$$
> where
> $$J_{meta,j,t}(\theta) =\mathrm{min}(\rho_{j,t}(\theta) A_{meta,j,t},\mathrm{clip}(\rho_{j,t}(\theta), 1-\epsilon_{low}, 1+\epsilon_{high}) A_{meta,j,t}).$$
>
> Please note that the advantages for the solution-generation prompt and the meta-prediction prompt are computed separately, since they use different prompts for different goals.
>
> ---
> ### **Point 2. Interaction between Meta-Prediction and Solution Rollout**
> Meta-prediction and solution paths interact only during training to improve meta-awareness and rollout efficiency.
> Since meta-predictions are not needed at inference, we use the model for standard CoT reasoning. However, explicitly leveraging meta-predictions to guide reasoning is a promising direction. We test this via notion feed-in (Table 3), inserting predicted notions as hints.
> We thank the reviewer for this insight and will include inference-time applications such as length regulation for efficiency and difficulty-based sampling to evaluate performance gains.
>
> ---
> ### **Point 3. How to Merge Rewards for Meta-Prediction**
> In our formulation, we used the unweighted sum because we view all three reward components as core aspects of meta-prediction, and we therefore chose the general aggregation strategy as our default design. We agree that using a weighted combination of the rewards could be an interesting direction for future work.
>
> ---
> ### **Point 4: Potential Use case of Meta-prediction**
> In our paper, we have leveraged the meta predictions for predictive gating and length cutoff for efficient rollout, and notion-based hinting for inference. Beyond these, the meta-predictions could be used for i) adaptive token length / sampling allocation during inference, ii) selective intervention on reasoning trace. We promise to incorporate the reviewer's valuable suggestion in the final copy of the paper.
>
> ---
> ### **Point 5: Candidate for Meta-Prediction Components**
> We agree that our framework could be extended beyond notion, score, and length to predict other forms of meta-information. One promising direction is to predict likely failure modes. By identifying the failure modes that are only present in the wrong reasoning trajectories, the model will be able to learn what are easy to fall pitfalls.
>
> ---
> ### **Point 6: Additional Statistics for Length Prediction and Predictive Gating**
> We analyze length and difficulty prediction reliability across benchmarks. Predicted difficulty strongly correlates with performance, and predicted vs. actual solution length also shows high agreement:
> | Metric                      | AIME2025 | AIME2024 | AMC23 |
> | - | - | -| - |
> | Difficulty vs. Performance  | 0.76     | 0.68     | 0.63  |
> | Predicted vs. Actual Length | 0.74        | 0.63        | 0.87     |
>
> However, length prediction degrades beyond the training budget. Trained with an 8192-token limit, the model becomes less reliable when extended to 16384 tokens on harder tasks like AIME, where outputs exceed the training range.
>
> We also observe that underestimating required length significantly reduces accuracy. On MATH500, adequate or overestimated budgets achieve >90% accuracy, while underestimated cases drop to 62.8%, highlighting the importance of accurate meta-prediction.
>
> ---
> ### **Point 7: Hypothesis on Quality of Solver and Final Performance**
> We agree that weaker solvers limit the benefit of meta-prediction.
> Our results in Table 5 support this. MAPR yields smaller gains for Llama3 and Gemma2 than for Qwen3, likely due to their lower base capability. This suggests a minimum solver quality is needed to produce reliable meta-predictions and realize performance gains.
>
> More broadly, MAPR is most effective for models with reasonable problem-solving ability, while weaker solvers may need improved reasoning before benefiting fully.

---

> > ### Author Rebuttal · Reviewer_JAqg · 2026-04-01
> >
> > My questions have been adequately answered. My misunderstanding on the use of MAPR-efficient is due to the potentials in using the authors' idea in inference, which is a confusion they will clarify.  I think the paper will improve with these changes and additions.

---

### Official Review · Reviewer_DMWT · 2026-03-11

**Soundness:** 3
**Presentation:** 3
**Significance:** 3
**Originality:** 3
**Overall Recommendation:** 4
**Confidence:** 4

**Summary:**

This paper introduces MAPR, which is a RL framework based on GRPO and it adds meta-awareness to reasoning models. During post-training, the model is trained to compute the meta-reward. To reduce computational cost, the authors also propose MAPR-efficient, which uses these predictions to skip unpromising rollouts and stop generation early. Experiments show that MAPR consistently outperforms the GRPO baseline in both accuracy and training efficiency.

**Compliance With Llm Reviewing Policy:**

Affirmed.

**Key Questions For Authors:**

1. How sensitive is the model's final performance to the choice of the reward base (0.01)?
2. Why does feeding the notions back into the prompt yield insignificant improvements?
3. In MAPR-efficient steps, does the strict early length cutoff stop the model from generating the longer, correct reasoning steps?

**Limitations:**

Yes

**Strengths And Weaknesses:**

## Strengths
1. Novel formulation of meta-awareness into quantifiable metrics for RL post-training, offering a verifiable approach to self-evaluation.
2. The authors evaluate their method on various foundation models and experimental results show that MAPR is Pareto-superior the baseline GRPO.
3. Insightful ablation studies, like the investigation of the drop-then-align training dynamics, which explain how the method corrects the model’s overconfidence.
4. MAPR successfully applies the meta-prediction results to avoid unnecessary token generation and it saves much computation during training.

## Weaknesses
1. It is unclear how the specific base value is chosen, and there is no sensitivity analysis provided.
2. The results for 'notion feed-in' (Table 3) are confusing. Feeding the notions back hardly improves accuracy, which makes me doubt if the model actually achieves meta-cognition.

---

> ### Author Rebuttal · Authors · 2026-03-31
>
> We sincerely appreciate the reviewer's effort in reviewing this paper. Below are the response for the reviewer's comment.
>
> ---
> ### **Point 1: Sensitivity Analysis on Base Number of Difficulty Reward**
> > The base number of 0.01 is meticulously set to halve by one unit of difference between the prediction and true difficulty. We set this parameter value without additional tuning, in accordance with the rationale described earlier. However, to address concerns regarding tuning, we conducted the following additional experiments. We provide additional results over different choice of base numbers (0.005, 0.01, 0.02). Therefore, the results are robust across different model sizes. Except for the extreme base number 0.005 for 4B model, which degrades the overall performance, all the other base number and model size combination shows robustness in final performance.
>
> | 4B | AIME24              | AIME25              | AMC23               | MATH500             | Minerva             | Olympiad            | Avg                 |
> |-------|--------------------|--------------------|---------------------|---------------------|---------------------|---------------------|---------------------|
> | 0.005 | 16.98 ± 3.30       | 14.58 ± 3.55       | 61.95 ± 5.33        | 79.62 ± 0.79        | **43.55 ± 1.89**    | 45.21 ± 0.91        | 43.65 ± 2.63        |
> | 0.01  | 26.15 ± 3.32       | 21.56 ± 4.40       | 70.16 ± 4.78        | 84.52 ± 0.74        | 41.12 ± 2.00        | **53.38 ± 0.96**    | 49.48 ± 2.70        |
> | 0.02  | **26.77 ± 4.62**   | **23.33 ± 3.97**   | **70.47 ± 5.74**    | **84.84 ± 0.99**    | 43.11 ± 1.98        | 53.24 ± 1.14        | **50.29 ± 3.07**    |
>
> | 8B | AIME24              | AIME25              | AMC23               | MATH500             | Minerva             | Olympiad            | Avg                 |
> |-------|--------------------|--------------------|---------------------|---------------------|---------------------|---------------------|---------------------|
> | 0.005 | **34.38 ± 5.66**   | 25.73 ± 3.55       | 78.05 ± 4.12        | **88.34 ± 0.92**    | **48.35 ± 1.48**    | **57.61 ± 1.08**    | 55.41 ± 2.80        |
> | 0.01  | 34.17 ± 5.54       | **28.44 ± 5.41**   | 79.53 ± 4.26        | 88.05 ± 0.82        | 47.21 ± 1.74        | 56.86 ± 0.85        | **55.71 ± 3.10**    |
> | 0.02  | 33.54 ± 6.55       | 24.17 ± 4.37       | **79.92 ± 4.16**    | 87.74 ± 0.76        | 46.19 ± 1.66        | 56.51 ± 0.87        | 54.68 ± 3.06        |
>
> ---
> ### **Point 2: Effect of Notion Feedin**
> > The notion feed-in results show limited performance improvement for our model MAPR as the notion keywords are internally learned by notion reward during training. Therefore the improvement is limited over the MAPR model without notion feedin. Therefore, we have conducted an experiment to test whether the extracted notions boost the performance of baseline GRPO model. Note that the notions are extracted from MAPR model, and fed into GRPO model by using keywords of high notion reward score.
>
> | Step     | AIME24        | AIME25        | AMC23         | MATH500       | Minerva       | Olympiad      | Avg            |
> |----------|--------------|--------------|--------------|--------------|--------------|--------------|----------------|
> | GRPO     | 28.54 ± 4.12 | 22.19 ± 3.63 | 73.67 ± 5.60 | 85.75 ± 0.66 | 43.21 ± 2.12 | 54.03 ± 1.22 | 51.23 ± 2.89  |
> | GRPO+NF  | 33.96 ± 5.88 | 23.85 ± 3.64 | 77.97 ± 5.08 | 86.52 ± 1.14 | 45.44 ± 1.54 | 56.63 ± 1.10 | 54.06 ± 3.06  |
> | MAPR     | 34.17 ± 5.54 | 28.44 ± 5.41 | 79.53 ± 4.26 | 88.05 ± 0.82 | 47.21 ± 2.12 | 56.86 ± 0.85 | 55.71 ± 3.10  |
>
> ---
> ### **Questions**
> **Q1**
> We have tested across three base numbers 0.005, 0.01, 0.02 with two different model sizes of Qwen3-4B and 8B in Point 1
>
> **Q2**
> We show the helpfulness of predicted meta notions for baseline GRPO model in Point 2.
>
> **Q3**
> In rollout sample level, 39.26% of cutoff rollout samples led to correct answer. However, in prompt level, the cutoff do not affect the zero-variance count, as it retains the same proportion of zero-variance prompts (41.4%) as the baseline that do not use length cutoff, indicating equal training signal. More importantly, it reduces rollout token usage by 48.93%, accelerating training.

---

> > ### Author Rebuttal · Reviewer_DMWT · 2026-04-03
> >
> > Thanks for the rebuttal. My concerns have been addressed.

---

### Official Review · Reviewer_Ncfd · 2026-03-12

**Soundness:** 3
**Presentation:** 3
**Significance:** 3
**Originality:** 3
**Overall Recommendation:** 4
**Confidence:** 3

**Summary:**

The paper incorporates auxiliary reward functions inspired by meta-awareness principle for reasoning RL training. Meta-awareness is instantiated through three dimensions: difficulty (pass@k), token length, and notion (keyword appearance). The key idea is simultaneously train the model for problem solving and meta-awareness using different groups of rollouts. They demonstrate that incorporating meta-awareness into training improves both the performance and training efficiency, producing more aligned model.

**Compliance With Llm Reviewing Policy:**

Affirmed.

**Final Justification:**

The authors provide additional results and justification which address my concern on robustness and generalization of the proposed methods.

**Key Questions For Authors:**

1. Can you provide more principled justification of the reward function design?
2. Can you explain how to determine whether a response contain a "notion"? Is it based on exact match? How robust is it?

**Limitations:**

yes

**Strengths And Weaknesses:**

Strengths:
1. Meta-awareness reward is derived from entirely self-supervision signals without requiring additional LLM-Judge / human intervention.
2. Notable performance gains on in-domain math benchmarks, also showing generalization to out-of-domain tasks.

Weaknesses:
1. Reward functions are designed based on heuristics. Some design choices are not well-motivated. For example, the base number of 0.01 in difficulty reward seems arbitrary; the notion reward may be brittle to lexical and semantic variation of the notion words.
2. It is unclear how effective the meta-awareness rewards would operate in other domains. The meta-prediction prompt in Appendix A is specifically designed for math questions. If training on other domains such as coding, the notion reward based on keyword appearance may not be applicable.

---

> ### Author Rebuttal · Authors · 2026-03-31
>
> We sincerely appreciate the reviewer's effort in reviewing this paper. Below is the response to the reviewer's comment.
>
> ---
> ### **Point 1: Additional Experiment on Hyper-parameter Choice**
> > ### **A. Base Number of Difficulty Reward**
> The base number of 0.01 is meticulously set to halve by one unit of difference between the prediction and true difficulty. We set this parameter value without additional tuning, in accordance with the rationale described earlier. However, to address concerns regarding tuning, we conducted the following additional experiments. We provide additional results over different choice of base numbers (0.005, 0.01, 0.02). Therefore, the results are robust across different model sizes. Except for the extreme base number 0.005 for 4B model, which degrades the overall performance, all the other base number and model size combination shows robustness in final performance.
>
> | 4B | AIME24              | AIME25              | AMC23               | MATH500             | Minerva             | Olympiad            | Avg                 |
> |-------|--------------------|--------------------|---------------------|---------------------|---------------------|---------------------|---------------------|
> | 0.005 | 16.98 ± 3.30       | 14.58 ± 3.55       | 61.95 ± 5.33        | 79.62 ± 0.79        | **43.55 ± 1.89**    | 45.21 ± 0.91        | 43.65 ± 2.63        |
> | 0.01  | 26.15 ± 3.32       | 21.56 ± 4.40       | 70.16 ± 4.78        | 84.52 ± 0.74        | 41.12 ± 2.00        | **53.38 ± 0.96**    | 49.48 ± 2.70        |
> | 0.02  | **26.77 ± 4.62**   | **23.33 ± 3.97**   | **70.47 ± 5.74**    | **84.84 ± 0.99**    | 43.11 ± 1.98        | 53.24 ± 1.14        | **50.29 ± 3.07**    |
>
> | 8B | AIME24              | AIME25              | AMC23               | MATH500             | Minerva             | Olympiad            | Avg                 |
> |-------|--------------------|--------------------|---------------------|---------------------|---------------------|---------------------|---------------------|
> | 0.005 | **34.38 ± 5.66**   | 25.73 ± 3.55       | 78.05 ± 4.12        | **88.34 ± 0.92**    | **48.35 ± 1.48**    | **57.61 ± 1.08**    | 55.41 ± 2.80        |
> | 0.01  | 34.17 ± 5.54       | **28.44 ± 5.41**   | 79.53 ± 4.26        | 88.05 ± 0.82        | 47.21 ± 1.74        | 56.86 ± 0.85        | **55.71 ± 3.10**    |
> | 0.02  | 33.54 ± 6.55       | 24.17 ± 4.37       | **79.92 ± 4.16**    | 87.74 ± 0.76        | 46.19 ± 1.66        | 56.51 ± 0.87        | 54.68 ± 3.06        |
>
> > ### **B. Variation in Notion**
> We preprocessed the keyword predictions using basic lemmatization and whitespace normalization, but did not account for lexical or semantic variations. A natural extension would be to leverage similarity in text embedding space. However, empirical observations suggest that mathematical terms show arbitrary proximity in embedding space, making such an approach unreliable. Additionally, incorporating rule-based parsing or LLM-as-a-judge mechanisms to handle these variations would introduce significant training-time overhead.
>
> ---
> ### **Point 2: Extension to Other Domains - Science and Coding**
>
> > Our notion-based reward is not inherently limited to mathematical domains. In coding tasks, the notion can be naturally mapped to the algorithmic components required for code generation and scientific terms for science QA domain. To verify this, we applied a simple prompt adaptation to a base model and observed that it can reliably extract meaningful notion predictions across domains.
>
> Below are the randomly selected results from sampling the meta-predictions from coding task LiveCodeBench.
>
> > 1. maximum-strength-of-a-group
> $\rightarrow$ ['array manipulation', 'mathematical operations', 'dynamic programming', 'greedy algorithms']
> 2. find-the-longest-equal-subarray
> $\rightarrow$  ['Sliding Window', 'Hash Map', 'Two Pointers']
> 3. greatest-common-divisor-traversal
> $\rightarrow$ ['graph traversal', 'gcd calculation', 'prime factorization']
>
> For science domain, we use GPQA Diamond and demonstrate several examples.
> > 1. Quantum mechanics problem
> $\rightarrow$ ['quantum mechanics', 'Heisenberg uncertainty principle']
> 2. Organic chemistry synthesis
> $\rightarrow$ ['organic chemistry', 'Grignard reactions', 'oxidation', 'reaction mechanisms']
> 3. Gene interaction problem
> $\rightarrow$ ['epistasis', 'transcription factor', 'gene redundancy']
>
> ---
> ### **Questions**
> **Q1** For difficulty reward, the base number 0.01 is set to halve the reward by one unit miss between prediction and ground truth. The notion reward incentivizes the notion present only in the correct ones, but not in incorrect. The length reward gives reward to correct prediction on the "correct" reasoning trace.
>
> **Q2** We have addressed this in Point 2-B.

---

> > ### Author Rebuttal · Reviewer_Ncfd · 2026-04-01
> >
> > My concerns have been adequately addressed.

---

> > > ### Author Response · Authors · 2026-04-02
> > >
> > > Dear Reviewer,
> > >
> > > Thank you again for your thoughtful evaluation and for indicating that your concerns have been fully resolved.
> > >
> > > We truly appreciate your positive assessment, and if you find that the additional experiments and clarifications sufficiently strengthen the paper, we would be very grateful if you could consider reflecting this in your final score.
> > >
> > > Thank you again for your time and support.
> > >
> > > Best regards,
> > >
> > > Authors

---

### Official Review · Reviewer_fdEN · 2026-03-12

**Soundness:** 2
**Presentation:** 3
**Significance:** 3
**Originality:** 2
**Overall Recommendation:** 3
**Confidence:** 4

**Summary:**

The paper introduces MAPR (Meta-Awareness via Predictive Reward), a reinforcement learning framework designed to improve reasoning models by incentivizing "meta-awareness." Instead of relying on external verifiers, MAPR tasks the model with predicting its own rollout statistics—specifically the expected pass rate, solution length, and necessary mathematical concepts. The predictions are then verified against the actual solution paths, creating a self-alignment reward. Furthermore, the authors propose an accelerated variant, MAPR-efficient, which leverages these meta-predictions to skip unsolvable prompts (Predictive Gating) and terminate overly long generations (Early Cutoff) . The authors demonstrate significant empirical gains on mathematical reasoning benchmarks, including an 83.18% relative improvement on AIME 25 compared to a GRPO baseline.

**Compliance With Llm Reviewing Policy:**

Affirmed.

**Final Justification:**

I have updated my score to 3.

**Key Questions For Authors:**

1.	Given that $r_{notion}$ accounts for 67.1% of the performance gain, how do you rule out the possibility of "shortcut learning," where the model merely learns to output a high volume of math keywords in the \<meta\> tag to guarantee a reward match without actually understanding them?
2.	What is the false-positive error rate for the Early Length Cutoff mechanism? Specifically, how often are potentially correct derivations prematurely terminated because the meta-prediction slightly underestimated the required tokens?
3.	The Predictive Gating uses a static online approach. Why was a dynamic or periodic re-evaluation not considered, and doesn't this static exclusion severely limit the model's ability to learn difficult concepts as its capabilities improve later in the epoch?
4.	How sensitive is the framework to the heuristic hyperparameters, specifically the exponential decay base of 0.01 for the difficulty reward and the start step $k=80$? Do these hold across different model sizes (e.g., 14B or 32B)?

**Limitations:**

See weaknesses.

**Strengths And Weaknesses:**

**Strengths:**

- **Significant Empirical Gains**: The reported performance improvements are substantial across major math benchmarks (e.g., AIME, AMC23, MATH500), proving the practical viability of the method .

- **Efficiency Improvements**: The MAPR-efficient variant successfully translates meta-awareness into actual computational savings, achieving baseline performance 1.28x faster than standard GRPO.
- **Insightful Ablation**: The use of Shapley $R^2$ analysis to quantify the exact contribution of each reward component (Difficulty, Length, Notion) is a rigorous methodological choice that provides excellent interpretability.

**Weaknesses:**

- **Severe Risk of Reward Hacking**: The Shapley analysis reveals that the "Notion Reward" ($r_{notion}$) dominates the performance gains, contributing an overwhelming 67.1%. This disproportionate reliance raises a critical red flag regarding "shortcut learning." The model may simply be learning to perform keyword matching—emitting high-frequency mathematical terms in its meta-trajectory to secure the reward —without genuinely deepening its logical reasoning.
- **Fragility of Efficient Mechanisms**: The efficiency gains rely on rigid, "hard" interventions. The Predictive Gating employs a "static online" approach; if the model predicts a prompt is unsolvable early in training, it is discarded and never re-evaluated , artificially capping the model's performance ceiling. Similarly, the Early Length Cutoff uses predicted length as a strict threshold . If underestimated, the model is physically prevented from reaching a correct answer, heavily penalizing valid Chain-of-Thought paths that simply required slightly more deliberation.
- **Over-reliance on Unjustified Heuristics**: Core components lack sensitivity analysis. The difficulty reward uses an aggressive exponential decay function with a base of 0.01. The transition to MAPR-efficient occurs exactly at step $k=80$ . These feel highly tuned to the specific 8B model and dataset, and it is doubtful they scale robustly without extensive retuning.
- **Missing Self-Correction Baselines**: While MAPR beats standard GRPO, the paper lacks a comparison against simpler self-consistency baselines (e.g., majority voting or a basic self-correction prompt) to prove that the complex dual-path RL framework is strictly necessary to achieve these filtering capabilities.

---

> ### Author Rebuttal · Authors · 2026-03-31
>
> We sincerely appreciate the reviewer's effort and time in reviewing this paper.
>
> ---
> ### **Point 1. MAPR do not have reward hacking issue**
> > The notion reward does not reward "frequent" keywords but discriminates "informative" keywords present only in correct but not wrong traces. This encourages understanding of problem context and correct reasoning. High-reward keywords include arc lengths, segments, cubic equation, lattice, triplets, while low-reward ones include appears, rule, run, possibility, code, string. This shows high-reward notions are problem-centric, while low-reward ones are generic. Results are collected across AMC23, AIME24, and AIME25 without cherry-picking. Moreover, the 67.1% contribution of notion reward is expected as it is a dense reward, unlike correctness / difficulty / length rewards.
>
> ---
> ### **Point 2. Robustness of Our Efficiency Mechanism**
> > ### **A. Static Online Approach is Necessary for Efficiency Objective**
> We use a static online approach for efficiency, as dynamic reuse of gated problems increases compute and conflicts with our objective. However, dynamically revisiting previously unsolvable problems can improve performance at additional cost.
> We verify this by showing that 15% of zero-variance prompts at step 100 become non-zero at step 300, indicating added learning signal. Thus, dynamic approaches help, but with increased compute.
>
> > ### **B. Length Cutoff do not hinder the Model Learning but Reduce Token Consumption**
> Length cutoff does not hinder reaching the performance ceiling. It retains the same proportion of zero-variance prompts (41.4%) as the baseline, indicating equal training signal. More importantly, it reduces rollout token usage by 48.93%, accelerating training. These results are observed at an intermediate step using DeepScaleR.
>
> ---
> ### **Point 3: Explanation on the Choices of Hyper-parameters**
> > ### **A. Base Number for Difficulty Reward**
> The base 0.01 halves the reward per unit difference between predicted and true difficulty, set without additional tuning based on prior rationale. To address tuning concerns, we evaluate bases {0.005, 0.01, 0.02}. Results are robust across model sizes and base numbers, except for extreme value of 0.005 on the 4B model, which degrades performance.
>
> | 4B | AIME24              | AIME25              | AMC23               | MATH500             | Minerva             | Olympiad            | Avg                 |
> |-------|--------------------|--------------------|---------------------|---------------------|---------------------|---------------------|---------------------|
> | 0.005 | 16.98       | 14.58       | 61.95        | 79.62        | **43.55**    | 45.21        | 43.65        |
> | 0.01  | 26.15       | 21.56       | 70.16        | 84.52        | 41.12       | **53.38**    | 49.48       |
> | 0.02  | **26.77**   | **23.33**   | **70.47**    | **84.84**    | 43.11        | 53.24        | **50.29**    |
>
> | 8B | AIME24              | AIME25              | AMC23               | MATH500             | Minerva             | Olympiad            | Avg                 |
> |-|-|-|-|-|-|-|-|
> | 0.005 | **34.38**   | 25.73       | 78.05        | **88.34**    | **48.35**    | **57.61**    | 55.41        |
> | 0.01  | 34.17       | **28.44**   | 79.53        | 88.05        | 47.21        | 56.86        | **55.71**    |
> | 0.02  | 33.54       | 24.17       | **79.92**    | 87.74        | 46.19        | 56.51        | 54.68        |
>
> > ### **B. Starting Point for Efficiency**
> Starting from step 0 achieves slightly better final performance than starting from step 80 for 14B model. However, all configurations converge to very similar final performance when starting from different start steps (around 45 in average).
>
> | Start Step | AIME24 | AIME25 | AMC23 | Avg |
> |-|-|-|-|-|
> | 0          | 35.83 | 26.04 | 75.78 | 45.88 |
> | 40         | 33.12 | 25.83 | 75.31 | 44.75 |
> | 80         | 34.27 | 26.98 | 75.94 | 45.73 |
> | 120        | 30.52 | 27.40 | 77.34 | 45.09 |
>
> ---
> ### **Point 4. Simpler Self-Consistency Method Fails to Learn Meta-Awareness**
> > We conduct two experiments following the reviewer’s suggestion to test self-consistency methods, specifically majority voting over meta-predictions as a learning signal. Majority voting shows very low reliability, especially in early training, leading to lost learning signals.
>
> | Difficulty (F1 / Precision / Recall) | Length (F1 / Precision / Recall) |
> |-|-|
> | 0.14 / 0.44 / 0.08             | 0.63 / 0.74 / 0.54         |
>
> ---
>
> ### **Questions**
> **Q1**: MARP predicts only 6.12 notions per problem, confirming no keyword over-generation.
>
> **Q2**: In rollout sample level, 39.26% of cutoff rollout samples led to correct answer. In prompt level, the cutoff do not affect the zero-variance count.
>
> **Q3**: Predictive gating improves efficiency, but periodic re-evaluation would improve performance but increases compute.
>
> **Q4**: Results across base values and start steps show robustness across model sizes.

---

> > ### Author Rebuttal · Reviewer_fdEN · 2026-04-03
> >
> > I have updated my score to 3.

---

> > > ### Author Response · Authors · 2026-04-04
> > >
> > > Dear Reviewer fdEN,
> > >
> > > Thank you very much for your thoughtful comments and for raising your score after reading our rebuttal.
> > > We sincerely appreciate your time and feedback.
> > >
> > > If there are any remaining concerns or suggestions, we would be grateful to learn them and further improve our work.
> > >
> > > Thank you again, and we hope you have a wonderful day!
> > >
> > > Best regards,
> > >
> > > Authors

---

### Decision · Program_Chairs · 2026-04-30

**Decision:**

Accept (regular)

**Comment:**

The paper incorporates Meta-Awareness via Predictive Reward for reasoning RL training. Meta-awareness include difficulty (pass@k), token length, and notion (keyword appearance). The key idea is simultaneously train the model for problem solving and meta-awareness using different groups of rollouts. The experimental results demonstrate significant empirical gains on mathematical reasoning benchmarks. According to the reviews, the main concerns are as follows.

1. The proposed meta-awareness may have the risk of reward hacking. The authors should consider the generalization problem of the proposed reward function.
2. The paper should make a comparison with simpler self-consistency baselines and verified the proposed method on more domains.